Pixel: a content management platform for quantitative omics data

Denecker Thomas 1
Durand William 2
Maupetit Julien 2
Hébert Charles 3
Camadro Jean-Michel 4
Poulain Pierre pierre.poulain@univ-paris-diderot.fr 4
Lelandais Gaëlle gaelle.lelandais@u-psud.fr 1
1 CEA, CNRS, Univ. Paris-Sud, Institute for Integrative Biology of the Cell (I2BC) , Gif-sur-Yvette , France
2 TailorDev SAS , Clermont-Ferrand , France
3 BIOROSETICS , Houilles , France
4 CNRS, Univ. Paris Diderot, Institut Jacques Monod (IJM) , Paris , France
Lazo Gerard
Electronic publication date: 2019 Mar 27
Publication date: 2019
Volume: 7
Electronic Location ID: e6623
Received 2018 Oct 5; Accepted 2019 Feb 14
Copyright: ©2019 Denecker et al.
Copyright year: 2019
Copyright holder: Denecker et al.
License: This is an open access article distributed under the terms of the Creative Commons Attribution License, which permits unrestricted use, distribution, reproduction and adaptation in any medium and for any purpose provided that it is properly attributed. For attribution, the original author(s), title, publication source (PeerJ) and either DOI or URL of the article must be cited.
License URL: https://creativecommons.org/licenses/by/4.0/

Keywords: Data cycle analyses, Omics, Open source, Pixel Web App

Funding: Agence Nationale pour la Recherche (CANDIHUB project ANR-14-CE14-0018-02 This work was funded by the Agence Nationale pour la Recherche (CANDIHUB project, grant number ANR-14-CE14-0018-02). The funders had no role in study design, data collection and analysis, decision to publish, or preparation of the manuscript.

==============================
Background

In biology, high-throughput experimental technologies, also referred as “omics” technologies, are increasingly used in research laboratories. Several thousands of gene expression measurements can be obtained in a single experiment. Researchers are routinely facing the challenge to annotate, store, explore and mine all the biological information they have at their disposal. We present here the Pixel web application (Pixel Web App), an original content management platform to help people involved in a multi-omics biological project.

Methods

The Pixel Web App is built with open source technologies and hosted on the collaborative development platform GitHub (https://github.com/Candihub/pixel). It is written in Python using the Django framework and stores all the data in a PostgreSQL database. It is developed in the open and licensed under the BSD 3-clause license. The Pixel Web App is also heavily tested with both unit and functional tests, a strong code coverage and continuous integration provided by CircleCI. To ease the development and the deployment of the Pixel Web App, Docker and Docker Compose are used to bundle the application as well as its dependencies.

Results

The Pixel Web App offers researchers an intuitive way to annotate, store, explore and mine their multi-omics results. It can be installed on a personal computer or on a server to fit the needs of many users. In addition, anyone can enhance the application to better suit their needs, either by contributing directly on GitHub (encouraged) or by extending Pixel on their own. The Pixel Web App does not provide any computational programs to analyze the data. Still, it helps to rapidly explore and mine existing results and holds a strategic position in the management of research data.

Introduction

In biology, high throughput (HT) experimental technologies—also referred as “omics”—are routinely used in an increasing number of research teams. Financial costs associated to HT experiments have been considerably reduced in the last decade (Hayden, 2014) and the trend in HT sequencing (HTS) is now to acquire benchtop machines designed for individual research laboratories (for instance Illumina NextSeq500 or Oxford Nanopore Technologies MinION, Blow, 2013). The number of HT applications in biology has grown so rapidly in the past decade that it is hard to not feel overwhelmed (Hadfield & Retief, 2018) (“The data deluge, 2012”). It seems possible to address in any organism, any biological question through an “omics” perspective, providing the right HT material and method are found. If HTS is often put at the forefront of “omics” technologies (essentially genomics and transcriptomics, Reuter, Spacek & Snyder, 2015), other technologies must be considered. Mass spectrometry (MS) for instance, enables HT identification and quantification of proteins (proteomics). Metabolomics and lipidomics are other derived applications of MS to characterize quantitative changes in small-molecular weight cellular components (Smith et al., 2014). Together, they all account for complementary “omics area” with the advantage to quantify distinct levels of cellular components (transcripts, proteins, metabolites, etc.).

Integration of datasets issued from different HT technologies (termed as multi-omics datasets) represents a challenging task from a statistical and methodological point of view (Huang, Chaudhary & Garmire, 2017). It implies the manipulation of two different types of data. The first type is the “primary data”, which correspond to raw experimental results. It can be FASTQ files for sequencing technology (Cock et al., 2010) or mzML files for MS (Martens et al., 2011). These files can be stored in public repositories such as SRA (Leinonen, Sugawara & Shumway, 2011), GEO (Clough & Barrett, 2016), PRIDE (Martens et al., 2005) or PeptideAtlas (Desiere et al., 2006). Analyses of primary data rely on standard bioinformatics protocols that for instance, perform quality controls, correct experimental bias or convert files from a specific format to another. A popular tool to analyse primary data is Galaxy (Afgan et al., 2016), which is an open web-based platform. “Secondary data” are produced upon analysis of primary data. It can be the counts of reads per genes for HTS results or the abundance values per proteins for MS results. In multi-omics datasets analysis, combining secondary data is essential to answer specific biological questions. It can be typically, the identification of differentially expressed genes (or proteins) between several cell growth conditions from transcriptomics (or proteomics) datasets, or the identification of cellular functions that are over-represented in a list of genes (or proteins). In that respect, secondary data can be analysed and re-analysed within a multitude of analytical strategies, introducing the idea of data analysis cycle. The researcher is thus constantly facing the challenge to properly annotate, store, explore and mine all the biological data he/she has at his/her disposal in a multi-omics project. This challenge is directly related to the ability to extract as much information as possible from the produced data, but also to the crucial question of doing reproducible research.

A Nature’s survey presented in 2016 indicates that more than 70% of the questioned researchers already experienced an impossibility to reproduce published results, and more than half of them were not able to reproduce their own experiments (Baker, 2016). This last point is intriguing. If experimental biology can be subjected to random fluctuations hardly difficult to control, computational biology should not. Running the same software on the same input data is expected to give the same results. In practice, replication in computational science is harder than people generally think (see Mesnard & Barba, 2017 as an illustration). It requires to adopt good practices for reproducible-research on a daily basis, and not only when the final results are about to be published. Initiatives to improve computational reproducibility exists (Peng, 2011; Stodden, Guo & Ma, 2013; Vasilevsky et al., 2017; Rougier et al., 2017; Stodden, Seiler & Ma, 2018), and today it is clear that the data alone are not enough to sustain scientific claims. Comments, explanations, software source codes and tests are prerequisites to ensure that an original research can be replicated by anyone, anytime, anywhere.

We developed the Pixel web application (Pixel Web App) with these ideas in mind. It is a content management platform to help the researchers involved in a multi-omics biological project, to collaboratively work with their HT data. The Pixel Web App does not store the primary data. It is rather focused on annotation, storage and exploration of secondary data (see Fig. 1). These explorations represent critical steps to answer biological questions and need to be carefully annotated and recorded to be further exploited in the context of new biological questions. The Pixel Web App helps the researcher to specify necessary information required to replicate multi-omics results. We added an original hierarchical system of tags, which allows to easily explore and select multi-omics results stored in the system and to use them for new interpretations. The Pixel Web App can be installed on any individual computer (for a single researcher for instance), or on a web server for collaborative work between several researchers or research teams. The entire software has been developed with high quality programming standards and complies to major rules of open-source development (Taschuk & Wilson, 2017). The Pixel project is available on GitHub at  https://github.com/Candihub/pixel, where full source code and detailed documentation are provided. We present in this article the Pixel Web App design and implementation. We provide a simple case study, emblematic of our daily use of the Pixel Web App, with the exploration of results issued from transcriptomics and proteomics experiments performed in the pathogenic yeast Candida glabrata.

Figure 1 Dataset flow through the Pixel Web App.

(A) Different types of datasets, which are managed in a multi-omics biological project. Primary and secondary datasets are two types of information arising from HT experimental technologies (see ‘Introduction’). Only secondary data and their associated Pixel Sets are stored in the Pixel Web App. Note that several Pixel Sets can emerge from multiple secondary data analyses. They comprise quantitative values (Value) together with quality scores (QS) for several hundred of different “Omics Units” elements (for instance mRNA or proteins, see the main text). Omics Units are identified with a unique identifier (ID). (B) Screenshot of the home page of the Pixel web interface. (C) Schematic representation of the data analysis cycles that surrounds the integration of Pixel Sets in the Pixel Web App (see the main text).

Material and Methods

Stack overview

The Pixel Web App provides researchers an intuitive way to annotate, store, explore and mine their secondary data analyses, in multi-omics biological projects. It is built upon mainstream open source technologies (see Fig. 2). Source code is hosted on the collaborative development platform GitHub (https://github.com/) and continuous integration is provided by CircleCI (https://circleci.com/). More precisely, the Pixel Web App uses the Python Django framework. This framework is based on a model-template-view architecture pattern, and data are stored in a PostgreSQL (https://www.postgresql.org/) database. We have built a docker image for the Pixel Web App. Other containers, Nginx (to serve the Django application) and PostgreSQL rely on official docker images. Each installation/deployment will result in the creation/execution of three docker instances: one for the Pixel Web App, one for the PostgreSQL database and one for the Nginx web server. In case of multiple installations, each trio of docker instances is fully isolated, meaning that data are not shared across multiple Pixel Web App installations.

Figure 2 Stack overview of the Pixel Web App.

Open source solutions used to develop Pixel are shown here. They are respectively used for the software development and test (blue section), the data storage (green section) and the web application for both staging and production (orange section).

Technical considerations

Docker images

The Pixel Web App is built on containerization paradigm (see Fig. 2). It relies on Docker (https://www.docker.com/),  i.e., a tool which packages an application and its dependencies in an image that will be run as a container. Docker helps developers to build self-contained images to run a software. These images are downloaded on the host system and used to build the Pixel Web App.

Minimal configuration and dependencies

The Pixel Web App can be deployed on Linux and MacOS operating systems (OS). Deployment on Windows is possible, but this situation will not be described here. Minimal requirements are: (i) 64 bits Unix-based OS (Linux/MacOS), (ii) Docker community edition >v18, (iii) Internet access (required in order to download the Docker images) and (iv) [optional] a web server (Apache or Nginx) configured as a reverse proxy.

Installation

A step-by-step tutorial to deploy the Pixel Web App can be found in the project repository (https://github.com/Candihub/pixel/blob/master/docs-install/how-to-install.md) together with a deploy script. To summarize, this script runs the following steps:

• Pull a tagged image of Pixel (web, see docker-composer file),

• Start all instances (web, db and proxy) recreating the proxy and web instances. Collect all static files from the Django app. These files will be served by the proxy instance.

• Migrate the database schema if needed (to preserve existing data).

Note that further technical considerations and full documentation can be found on GitHub repository associated to the Pixel project (https://github.com/Candihub/pixel/tree/master/docs).

Results

Definition of terms: Omics Unit, Pixel and Pixel Set

In the Pixel Web App, the term “Omics Unit” refers to any cellular component, from any organism, which is of interest for the user. The type of Omics Unit depends on the HT experimental technology (transcriptomic, proteomic, metabolomic, etc.) from which primary and secondary datasets were collected and derived (Fig. 1A). In this context, classical Omics Units can be transcripts or proteins, but any other cellular component can be defined as, for instance, genomic regions with “peaks” in case of ChIPseq data analyses (Merhej et al., 2014). A “Pixel” refers to a quantitative measurement of a cellular activity associated to a single Omics Unit, together with a quality score (see Fig. 1A). Quantitative measurement and quality score are results of statistical analyses performed on secondary datasets, e.g., search for differentially expressed genes (Seyednasrollah, Laiho & Elo, 2015). A set of Pixels obtained from a single secondary data analysis of HT experimental results is referred as a “Pixel Set” (see Fig. 1A). Pixel Sets represent the central information in the Pixel Web App and functionalities to annotate, store, explore and mine multi-omics biological data were designed according to this concept (see below).

Functionalities to annotate, store, explore and mine Pixel Sets

Pixel Sets are obtained from secondary data analyses (see Fig. 1A). Their manipulation with the Pixel Web App consists in (i) their annotation, (ii) their storage in a database, (iii) their exploration and (iv) their mining (see Fig. 1C). This represents a cycle of multiple data analyses, which is essential in any multi-omics biological project. These different steps are detailed in the following.

Figure 3 Data modelling in the Pixel Web App.

The Pixel Set is the central information (see Fig. 1A), the corresponding table in the model is highlighted in red. Information that is required before Pixel Set import in the Pixel Web App is surrounded in blue, whereas information required during Pixel Set import is highlighted in orange. Other tables are automatically updated during the Pixel Web App data analysis life cycle (see Fig. 1C). An enlarged version of this picture together with full documentation is available online https://github.com/Candihub/pixel/blob/master/docs/pixel-db.pdf.

Annotation of Pixel Sets

Annotation of Pixel Sets consists in tracking important details of Pixel Set production. For that, Pixel Sets are associated with metadata, i.e., Supplemental Information linked to the Pixel Sets. We defined minimal information necessary for relevant annotations of Pixel Sets (see Fig. 3). “Species”, “Strain”, “Omics Unit Type” and “Omics Area” are mandatory information that must be specified before a new Pixel Set submission (highlighted in blue, Fig. 3). They refer to general information related to the multi-omics biological project on which the researcher is working on: (i) the studied organism and its genetic background (Species and Strain, e.g., Candida glabrata and ATCC2001), (ii) the type of monitored cellular components (Omics Unit Type, e.g., mRNA, protein) and (iii) the nature of the experimental HT technology (Omics Area, e.g., RNA sequencing, mass spectrometry). All Omics Units must be declared in the Pixel Web App before new Pixel Set submission. They must be defined with a short description and a link to a reference database. “Experiment” and “Analysis” are Pixel Set mandatory information, input during the submission of new Pixel Sets in the Pixel Web App (highlighted in orange, Fig. 3). They include, respectively, the detailed description of the experimental strategy that was applied to generate primary and secondary data sets (Experiment) and the detailed description of the computational procedures that were applied to obtain Pixel Sets from secondary data set (Analysis). Information regarding the researcher who performed the analyses is referred as “Pixeler”.

Storage of Pixel Sets in the database

Import of new Pixel Sets in the Pixel Web App requires the user to follow a workflow for data submission. It corresponds to six successive steps that are explained below (Fig. 4A).

Figure 4 Procedure to import new Pixel Sets in the Pixel Web App.

(A) New data-sets are submitted following a dedicated workflow that comprised 6 successive actions named “Download”, “Upload”, “Meta”, “Validation”, “Tags” and “Import archive” (see 1). Several files are required (see 2): the secondary data from which the Pixel Sets were calculated, the notebook in which the procedure to compute Pixel Sets from secondary data is described and the Pixel Set files (2 files in this example). A progression bar allows the user to follow the sequence of the submission process. (B) Excel spreadsheet in which annotations of Pixel Sets are written. Information related to the Experiment (see 1), the Analysis (see 2) and the Pixel datasets (see 3) is required. Note that this file must be downloaded at the first step of the submission process (“Download”, see A), allowing several cells to be pre-filled with annotations stored in the database (see 4 as an illustration, with Omics area information). (C) All information filled in the Excel file (see B) is extracted and can be modified anytime through a dedicated web page as shown here. User can edit the Pixel Set (see 1), edit the analysis(see 2), edit the experiment (see 3) and add “Tags” (see 4). The Tags are of interest to further explore Pixel Sets in the Pixel Web App.

1. The “Download” step consists in downloading a template Excel file from the Pixel Web App (see Fig. 4B). In this file, multiple-choice selections are proposed for “Species”, “Strain”, “Omics Unit Type” and “Omics Area” fields. These choices reflect what is currently available in the database and can be easily expanded. User must fill other annotation fields related to the “Experiment”, “Analysis” and “Pixeler” information. The Excel file is next bundled into a ZIP archive with the secondary data file (in tab-separated values format), the user notebook (R markdown (https://rmarkdown.rstudio.com/) or Jupyter notebook (http://jupyter.org/) for instance) that contains the code used to produce the Pixel Sets from the secondary data file.

2. The “Upload” step consists in uploading the ZIP file in the Pixel Web App.

3. The step “Meta” consists in running an automatic check of the imported file integrity (md5sum checks are performed, Excel file version is verified, etc.). Note that no information is imported in the database at this stage, but a careful inspection of all Omics Units listed in the submitted Pixel Sets is done. This is why Omics Units need to be pre-registered in the Pixel Web App (see previous section).

4. In “Annotation” step, the annotations of Pixel Sets found in the Excel file (see Fig. 4C) are controlled and validated by the user.

5. Next, the “Tags” step is optional. It gives the opportunity to the user to add tags to the new Pixel Sets (see Fig. 4C), that could be helpful for further Pixel Set explorations (see next section).

6. The final step “Import archive” consists in importing all Pixel Sets in the database, together with annotations and tags.

Note that the procedure of importing metadata as an Excel file has been inspired from the import procedure widely used in GEO (Clough & Barrett, 2016).

Exploration of Pixel Sets

The Pixel Web App aims to help researchers to mine and integrate multiple Pixel Sets stored in the system. We developed a dedicated web interface to explore all the Pixel Sets stored in a particular Pixel instance (see Fig. 5).  The upper part named “Selection” lists a group of Pixel Sets selected by the user for further explorations (Fig. 5A). The middle part named “Filters” lists the Pixel database contents regarding the Species, Omics Unit Types, Omics Areas and Tags annotation fields. The user can select information (Candida glabrata and modified pH here), search and filter the Pixel Sets stored in the database (Fig. 5B). The lower part is a more flexible search field in which keywords can be type. These keywords are searched in the Analysis and Experiment detailed description fields as illustrated here with LIMMA. The web interface also comprised detailed information for the selected subset of Pixel Sets with for instance, distributions of values and quality scores and a list of individual Omics Unit shown at the bottom of the page (Fig. 5C). Note that tags have been implemented to offer to the user a versatile yet robust annotation of Pixel Sets. They are defined during the import process, but they can be modified at any time through the Pixel web interface. Once searched, matching Pixel Sets are gathered in a table that can be exported.

Figure 5 Functionalities to explore the Pixel Sets stored in the Pixel Web App.

(A) Screenshot of the exploration menu available via the web interface. (B) Screenshot of the table that comprises all Pixel Sets, which match the filter criteria (see A). Particular Pixel Sets can be selected here (for instance “Pixel_C10.txt” and “Pixel_C60.txt”). They will therefore appear in the “Selection” list (see A). (C) Screenshot of the web interface that gives detailed information for the selected subset of Pixel Sets (see A). Distribution of values and quality scores are shown and individual Omics Unit are listed at the bottom of the page.

A case study in the pathogenic yeast Candida glabrata

The yeast Candida glabrata (C. glabrata) is a fungal pathogen of human (Bolotin-Fukuhara & Fairhead, 2014). It has been reported as the second most frequent cause of invasive infections due to Candida species, i.e., candidemia, arising especially in patients with compromised immunity (HIV virus infection, cancer treatment, organ transplantation, etc.). Candidemia remains a major cause of morbidity and mortality in the healthcare structures (Horn et al., 2009; Pfaller et al., 2012). The genome of Candida glabrata has been published in 2004 (Dujon et al., 2004). Its size is 12.3 Mb with 13 chromosomes and is composed of ∼5,200 coding regions.  Our research team is familiar with functional genomic studies in C. glabrata. In collaboration with experimental biologists, we published in the past ten years half dozen of articles, in which HT technologies were used  (Lelandais et al., 2008; Goudot et al., 2011; Merhej et al., 2015; Merhej et al., 2016; Thiébaut et al., 2017). In our lab, the Pixel Web App is installed locally and store all the necessary genomics annotations to manage any multi-omics datasets in this species.

As a case study, we decided to present how the Pixel Web App can be helpful to answer a specific biological question with only a few mouse clicks. As a biological question, we wanted to identify the genes in the entire C. glabrata genome: (i) which are annotated as involved in the yeast pathogenicity and (ii) for which the expression is significantly modified in response to an environmental stress induced by alkaline pH. Indeed, during a human host infection, C. glabrata has to face important pH fluctuations (see Ullah et al., 2013; Brunke & Hube, 2013; Linde et al., 2015 for more detailed information). Understanding the molecular processes that allow the pathogenic yeast C. glabrata to adapt extreme pH situations is therefore of medical interest to better understand host-pathogen interaction (Linde et al., 2015).

In a paper published in 2015, Linde et al. (2015) provided a detailed RNAseq based analysis of the transcriptional landscape of C. glabrata in several growth conditions, including pH shift experiments.  The primary dataset (RNAseq fastq files) is available in the Gene Expression Omnibus (Clough & Barrett, 2016) under accession number GSE61606. The secondary dataset (log2 Fold Change values) is available in Table S1 on the journal website (https://academic.oup.com/nar/article/43/3/1392/2411170). A first Pixel Set (labelled A) was created from this secondary dataset, annotated and imported into our Pixel Web App instance, following the procedure previously described. The associated ZIP archive is provided as Supplemental Information, along with all the details related to the experiment set up and the analysis. The Pixel Set A thus illustrates how publicly available data can be managed with the Pixel Web App. In our laboratory, we performed mass spectrometry experiments that also include pH shift (ZIP archive of the data is provided as Supplemental Information). Secondary dataset issued from these experiments leads to the Pixel Set B. Pixel Sets A and B comprise 5,253 Pixels and 1,879 Pixels (Fig. 6).

Figure 6 Case study in the pathogenic yeast Candida glabrata.

Our Pixel Web App was explored with the keywords “Candida glabrata” and “alkaline pH”. Two Pixel Sets were thus identified because of their tags. Two other tags were identical between the two Pixel Sets (“WT” and “logFC”), indicating that (i) C. glabrata strains are the same, i.e., Wild Type, and (ii) Pixel values are of the same type, i.e., log Fold Change. Notably Pixel Set A is based on transcriptomics experiments (RNAseq, see the main text), whereas Pixel Set B is based on proteomics experiments (mass spectrometry, see the main text). Omics Unit were next explored using the keyword “pathogenesis” resulting in the identification of 17 Pixels (respectively 6 Pixels) in transcriptomics (respectively proteomics) results. They were combined and exported from the Pixel Web App, hence starting a new data analysis cycle.

Transcriptomics (Pixel Set A) and proteomics (Pixel Set B) are interesting complementary multi-omics information that can be easily associated and compared with the Pixel Web App. In that respect, tags allowed to rapidly retrieve them using the web interface, applying the keywords “Candida glabrata” and “alkaline pH” (Fig. 6, Step 1). As we wanted to limit the analysis to the C. glabrata genes potentially involved in the yeast pathogenesis, a filter could be used to only retain the Omics Units for which the keyword “pathogenicity” is written in their description field (see Fig. 6, Step 2). As a result, a few numbers of Pixels were thus selected, respectively 17 in Pixel Set A and 6 in Pixel Set B. The last step consists in combining the mRNA and protein information (see Fig. 6, Step 3). For that a table comprising the multi-pixel sets can be automatically generated and easily exported. We present Table 1: five genes for which logFC values were obtained both at the mRNA and the protein levels, and for which statistical p-values were significant (<0.05). Notably two genes (CAGL0I02970g and CAGL0L08448g, lines 3 and 5 in Table 1) exhibited opposite logFC values, i.e., induction was observed at the mRNA level whereas repression was observed at the protein levels. Such observations can arise from post-translational regulation processes or from possible experimental noise, which could explain approximative mRNA or protein quantifications. In both cases, further experimental investigations are required. The three other genes (CAGL0F04807g, CAGL0F06457g and CAGL0I10516g, underlined in grey in Table 1) exhibited multi-omics coherent results and significant inductions were observed at the mRNA and protein levels. Again, further experimental investigations are required to fully validated these observations. Still, it is worth noting that the gene CAGL0F04807g, is described as “uncharacterized” in the Candida Genome Database (http://www.candidagenome.org/cgi-bin/locus.pl?locus=CAGL0F04807g&organism=C_glabrata_CBS138). Considering that logFC values for this gene are particularly high (>1), such an observation represents a good starting point to refine the functional annotation of this gene, clearly supporting the hypothesis that it has a role in the ability of C. glabrata to deal with varying pH situations.

Table 1 Detailed information regarding the Omics Unit identified in the C. glabrata case study.

The two first columns give Omics Unit information as described in the Candida Genome Database (Skrzypek et al., 2017). All the description fields comprise the keyword “pathogenesis” (in bold). LogFC values measured in transcriptomic (Pixel Set A) and proteomic (Pixel Set B) experiments are shown in the third and fourth columns. Quality scores (QS) are following logFC values. They are p-values coming from the differential analysis of logFC replicates. The entire table of multi-pixel sets is available in Supplementary Data.

Omics unit	Description	A	B	A (QS)	B (QS)	
1. CAGL0F04807g	Ortholog(s) have role in pathogenesis and cell surface, hyphal cell wall, integral component of mitochondrial outer membrane, plasma membrane localization	1,09	1,81	2,23E−19	7,31E−05	
2. CAGL0F06457g	Ortholog(s) have role in fungal-type cell wall organization or biogenesis, mitochondrial outer membrane translocase complex assembly, pathogenesis, phospholipid transport, protein import into mitochondrial outer membrane	0,30	0,19	4,14E−02	2,65E−01	
3. CAGL0I02970g	Ortholog(s) have delta14-sterol reductase activity and role in cellular response to drug, ergosterol biosynthetic process, filamentous growth of a population of unicellular organisms in response to biotic stimulus, pathogenesis	0,90	−2,64	4,65E−16	2,19E−05	
4. CAGL0I10516g	Ortholog(s) have role in fungal-type cell wall organization, pathogenesis and cytoplasm, eisosome, integral component of plasma membrane, membrane raft localization	1,50	0,57	8,29E−60	1,16E−02	
5. CAGL0L08448g	Ortholog(s) have role in actin cytoskeleton organization, eisosome assembly, negative regulation of protein phosphorylation, negative regulation of sphingolipid biosynthetic process and pathogenesis	1,67	−0,57	1,77E−75	7,04E−03	

Software availability

Pixel is released under the open-source 3-Clause BSD license (https://opensource.org/licenses/BSD-3-Clause). Its source code can be freely downloaded from the GitHub repository of the project: https://github.com/Candihub/pixel. In addition, the present version of Pixel (4.0.4) is also archived in the digital repository Zenodo (https://doi.org/10.5281/zenodo.1434316).

Discussion

In this article, we introduced the principle and the main functionalities of the Pixel Web App. With this application, our aim was to develop a tool to support on a daily basis, the biological data mining in our multi-omics research projects. It is our experience that research studies in which HT experimental strategies are applied, require much more time to analyse and interpret the data, than to experimentally generate the data. Testing multiple bioinformatics tools and statistical approaches is a critical step to fully understand the meaning of a biological dataset and in this context, the annotation, the storage and the ability to easily explore all results obtained in a laboratory can be the decisive steps to the success of the entire multi-omics project.

The data modelling around which the Pixel Web App was developed has been conceived to find a compromise between a too detailed and precise description of the data (which could discourage the researchers from systematically using the application after each of their analyses) and a too short and approximate description of the data (which could prevent the perfect reproduction of the results by anyone). Also, attention has been paid to allow heterogeneous data, i.e., different Omics Unit Type quantified in different Omics Area, to be stored in a coherent and flexible way. The Pixel Web App does not provide any computational programs to analyse the data. Still, it allows to explore existing results in a laboratory and to rapidly combine them for further investigations (using for instance the Galaxy platform or any other data analysis tool).

Therefore, the Pixel Web App holds a strategic position in the data management in a research laboratory, i.e., as the starting point but also at the final point of all new data explorations. It also helps data analysis reproducibility and gives a constant feedback regarding the frequency of the data analysis cycles; the nature of the import and export data sets as well as full associated annotations. It is thus expected that the content of different Pixel Web App instance will evolve with time, according to the type of information stored in the system and the scientific interests of a research team.

Conclusion

The Pixel Web App is freely available to any interested parties. The initial installation on a personal workstation required IT support from a bioinformatician, but once this is done, all administration tasks can be performed through the Web Interface. This is of interest for user with a few technical skills. We chose to work exclusively with open source technologies and our GitHub repository is publicly accessible (https://github.com/Candihub/pixel). We thus hope that the overall quality of the Pixel Web App source code and documentation will be guaranteed over time, through the shared contributions of other developers.

Supplemental Information

Supplemental Information 1 Zip archive associated to the PixelSet A

Mutli-omics data to be imported in the Pixel Web App to reproduce the case study presented in the main text.

Click here for additional data file.

Supplemental Information 2 Zip archive associated to the PixelSet B

Mutli-omics data to be imported in the Pixel Web App to reproduce the case study presented in the main text.

Click here for additional data file.

Additional Information and Declarations

Competing Interests

Author Contributions

Data Availability

William Durand and Julien Maupetit are employed by TailorDev SAS. Charles Hébert is employed by Biorosetics.

Thomas Denecker performed the experiments, contributed reagents/materials/analysis tools, prepared figures and/or tables, authored or reviewed drafts of the paper, approved the final draft.

William Durand conceived and designed the experiments, performed the experiments, approved the final draft.

Julien Maupetit and Charles Hébert conceived and designed the experiments, performed the experiments, contributed reagents/materials/analysis tools, approved the final draft.

Jean-Michel Camadro analyzed the data, approved the final draft.

Pierre Poulain conceived and designed the experiments, contributed reagents/materials/analysis tools, authored or reviewed drafts of the paper, approved the final draft.

Gaëlle Lelandais conceived and designed the experiments, analyzed the data, contributed reagents/materials/analysis tools, prepared figures and/or tables, authored or reviewed drafts of the paper, approved the final draft.

The following information was supplied regarding data availability:

Source code can be freely downloaded from the GitHub repository of the project: https://github.com/Candihub/pixel.

The present version of Pixel (4.0.4) is also archived at Zenodo:

Durand, William, Maupetit, Julien, Denecker, Thomas, Hébert, Charles, Poulain, Pierre, & Lelandais, Gaëlle. (2018, September 24). Pixel (v4.0.4): Integration of smart ‘omics’ data (Version 4.0.4). Zenodo. http://doi.org/10.5281/zenodo.1434316.

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
