# Peer review of "Pixel: a content management platform for quantitative omics data"

_PeerJ, doi:10.7717/peerj.6623_

## Round 0.1 · original submission · Major Revisions

In general the manuscript was well received; however, with some question of the claims stated, and the use case examples presented to demonstrate a test-run of the software. With the newer ways of testing software, might I suggest the development of some sort of DOCKER container that might be used as a test platform for a potential user to test before installing. The reviewers present some valid use case scenarios which are not apparently addressed within the scope of the manuscript; perhaps expanding a number of test-sets would be of value. As another way of demonstrating the value of the software, an example of a “deeper” set of data to demonstrate the utility with high-throughput “big” data types might be formulated as an example. With claims compared against Galaxy, the actual benchmark would really need to be measured. The reviewers provide some valid suggestions which should be addressed. The feedback provided should provide a valuable starting point for strengthening the presentation of the software. Since there may be considerable revision to address the suggested points I will rank this as requiring “Major Revision”, but it may be easier to adjust than the categorization implies. We look forward to your revision.

·

Basic reporting

The manuscript is overall well-written, figures are clear and easy to follow. Although I like the general idea of organising secondary data more systematically, I was not fully convinced what Pixel’s USP was compared to popular and well-established tools like Galaxy which support many data types and offer data tagging.

1. Title is misleading. "Digital lab assistant" can mean anything. "Multi-omics": Pixel is made for a very specific type of omics data. There are many types of omics data that Pixel does not support, e.g. protein-protein interaction data. Similarly, the term data integration is misleading, as it usually implies the utility of ontologies, standard vocabularies, URI to integrate data - none of which are used in Pixel as far as I can tell. I suggest to use a different title, e.g.: “Pixel - a content management platform for quantitative omics data”.

2. Galaxy comparison. The latest version of Galaxy provides tags and group tags (https://docs.galaxyproject.org/en/release_18.09/releases/18.09_announce.html). Primary data can be tagged, and tags are propagated all the way to secondary data in Galaxy histories/ libraries. Galaxy has tools to filter, join and visualise tables. I like to know what the USP of Pixel is compared to Galaxy?

Experimental design

The software stack used to develop and deploy the software is cutting edge. The presented use case nicely illustrates the functionalities of Pixel. First, filters are used to identify two datasets tagged with the same species and “alkaline pH”. Second, genes/proteins are filtered if they contain “pathogenesis” in their description. It was not clear in Line 259-262 where the description came from.

My concerns are:
3. Reproducibility. Pixel clearly helps with data/content management, but I can’t figure out how it helps with reproducibility. As the authors clearly state, Pixel is not a tool for primary data analysis, it is for managing and visualising secondary data. When users upload data into Pixel, they can provide a description of the “Analysis” and “Experiment”. It is very optimistic to expect users would provide tool versions and parameters that created the secondary data. I would recommend to remove any references to reproducibility.

4. Metadata and Tags. The success of Pixel’s search depends on the richness and consistency of user-provided tags and metadata. Two users could choose different tags to describe the same dataset. How does Pixel cope with situations when different terms are used to describe the similar data, or with spelling mistakes?

5. Use case. It would be good if the author’s can illustrate why doing the same task in Galaxy would be more difficult: tag search, text search, join three tables and histogram visualisation.

6. Excel. Despite its universality it is notorious for identifier mangling (https://genomebiology.biomedcentral.com/articles/10.1186/s13059-016-1044-7). I would like to learn what safeguards there are against this.

Validity of the findings

Minor technical issues:
I got this following the README:
- ‘make bootsrap’ (first step):
- django.db.utils.ProgrammingError: relation "core_omicsarea" does not exist
Went away after re-running the command
- Then I got: /bin/sh: yarn: command not found
- after installing 'yarn', it went fine

The step 'make dev', last message from 'make dev' is 'Creating pixel-dev_node_1 ... done'. At this point the server seems started, but the stdout isn't clear on this.
After this step, I can see a login page at http://127.0.0.1:8000, but I have no clue what to do from there.
There are requirements like Yarn, which the documentation should mention.
The documentation should say something about what to do once the application is deployed (is there a default user? Should I create it) and it should mention first steps I can do as user.
Moreover, regarding the step 'or run the Django development server and the css build watcher separately', it's not clear what to do as initial steps in either case (clone and cd).

·

Basic reporting

Denecker et al. describe the Pixel infrastructure, a digital lab assistant to manage, annotate and integrate datasets produced with a variety of omics technologies. In the first part of the manuscript, the authors describe the software in terms of underlying technology and design, and provide and overview of its usage. They then provide a short use case, leading to the identification of units potentially involved in the pathogenicity of Candida glabrata under alkaline stress using two omics datasets. The article is generally well written and conveys a good idea of the software's capabilities and the underlying technologies and development model.

In terms of spelling, I have two comments

- Line 79, possibly replace Importation with Import.
- Line 255, 'are search_ed_'

Experimental design

Here, I will comment on the software and its features:

- I struggled to install the software locally. Given the open development model used by the team, I will directly address this in a GitHub issue.

- As the authors acknowledge, installation of the software would require dedicated support, even if its use, later, is very intuitive. Given my own issues to install and test Pixel, and that potential users would want to try it out before installing it locally, it would be useful if a public instance could be used as test bench.

- I think further clarification or examples of what the Value and QS can be used for would be helpful. It becomes clear, later, in the example, that they can be used to store fold changes and p-values, but can any other data be used such as, for example classification or clustering results? Is there thus a limitation to 2 quantitative values per unit? How would one deal with a case where the experiment compared more than two groups and more fold changes and p-values are to be reported?

- Are the tags used to document Pixel sets free text, or is there some controlled vocabulary that will prevent different users to use different synonyms for the same term?

- On lines 188-190, I think there's a need for more information about 'user notebooks'. Are these dynamically executed scripts, or only provided as a documentation? Is this optional or mandatory?

- I would suggest to link to documentation in docs and docs-install in README.md file.

- Is it possible to export and update Pixel sets?

- On line 315, at the end of the discussion, the authors explain that a private web app contains 20000 pixels. Following the definition of a pixel being one line/entry in a pixel set, this seems very low.

Validity of the findings

Here, I will focus on the use case.

My main concern is that Pixel Set A zip file is missing and that the Pixel Set B isn't available at all. Hence, even if I had managed to install the software, I wouldn't be able to test it by reproducing the use case presented in the manuscript.

I also find the use case very simplistic, or lacking details, to make a convincing case. For example, the integration, based on common identifiers (unit names), which I assume, need to be mapped accordingly prior to importing the data into Pixel, is trivial. How would Pixel deal with metabolomics data, where liking between gene/protein names and metabolites isn't as direct? A more elaborated use case, that would illustrate a data analysis cycle as presented in figure 1C, would be useful.

In the use case, units matching the 'pathogenicity' term are searched for. How were these units annotated originally? This seems to be crucial to be able to perform a search and identify the units of interest. How could a user identify units matching a specific GO term, for example?

In the use case, the authors use p-values as an example. Are these not adjusted p-values, or shouldn't they be? This, in my opinion, also reduces the relevance of the use case.

Additional comments

To summarise, my main concerns are

- being able to test the software and either reproduce the use case, or at least use test data provided by the developers;
- the use case could be improved to make a stronger case in favour of Pixel.

---

## Round 0.2 · accepted · Accept

Thank you for the close attention to detail expressed by the reviewers. I have considered the revision - the manuscript appears well refined and is ready to be moved forward. The software should have utility to many users in the high-throughput data arena. Consider your manuscript Accepted for Publication. Thank you for your contribution.

#

---

## Author Rebuttal · Round 0.2

# Editor's Decision

*In general the manuscript was well received; however, with some question of the claims stated, and the use case examples presented to demonstrate a test-run of the software. With the newer ways of testing software, might I suggest the development of some sort of DOCKER container that might be used as a test platform for a potential user to test before installing. The reviewers present some valid use case scenarios which are not apparently addressed within the scope of the manuscript; perhaps expanding a number of test-sets would be of value. As another way of demonstrating the value of the software, an example of a "deeper" set of data to demonstrate the utility with high-throughput "big" data types might be formulated as an example. With claims compared against Galaxy, the actual benchmark would really need to be measured. The reviewers provide some valid suggestions which should be addressed. The feedback provided should provide a valuable starting point for strengthening the presentation of the software. Since there may be considerable revision to address the suggested points I will rank this as requiring "Major Revision", but it may be easier to adjust than the categorization implies. We look forward to your revision.*

Author answer:

Dear editor,

First of all we would like to thank both reviewers for their careful reading of our manuscript, for their positive comments and for their relevant remarks. We also would like to apologize for the delay we took to answer and submit a revised version of our article. We have teaching responsibilities at the University and during the last two months, they have been very time consuming. We thank you, and the reviewers for your understanding.

You can find below point-by-point answers to all the remarks made by the two reviewers. We hope our explanations and the modifications we made in the main text, will clarify the interest of the  Pixel Web App for other research teams facing the challenge to deal with multi-omics datasets.

Gaëlle Lelandais & Pierre Poulain
* * *
# Reviewer: Keywan Hassani-Pak

## Basic reporting

*The manuscript is overall well-written, figures are clear and easy to follow. Although I like the general idea of organising secondary data more systematically, I was not fully convinced what Pixel's USP was compared to popular and well-established tools like Galaxy which support many data types and offer data tagging.*

Author answer:

The reviewer is right. The positioning of our Pixel Web Application, with respect to the Galaxy online platform, needed to be clarified.

Very popular, the Galaxy platform became a standard for bioinformatics analyses of "omics" data. When multi-omics data started to accumulate in our laboratory, we decided to install a local instance on our server (this instance was functional between 2014 and 2017). With this Galaxy instance, our aim was *i*) to store all primary and secondary datasets we add with appropriate annotations, *ii*) to share them between our team members, and *iii*) to write and register workflows for all analyses performed on primary and secondary datasets. In this context, we rapidly experienced several technical problems, in relation essentially with *i*) the procedure to share data between users *via* the Galaxy "history" system and *ii*) our ability to manage the backend administration of this Galaxy local instance.

As an illustration, we observed that the sharing of histories between users was associated with an entire duplication of the datasets on our laboratory server, with the undesirable consequences of disk space and performance issue. Different users were thus working on duplicated primary and secondary data files. We understand that this makes sense in the Galaxy philosophy, but in our laboratory, this was clearly not an optimal solution for data sharing. Another difficulty we encountered was related to the procedures to clean deleted histories, which were (at this time) possible only via the execution of Python scripts on the server (we observed that the procedure to delete history from the web interface was only partial). Due to lack of storage space on our server, we had to apply these maintenance operations relatively often, requiring each time the intervention of our system administrator. One alternative solution for us would have been to use a public Galaxy instance (for example https://usegalaxy.eu/ or https://usegalaxy.org/) but we were worried about not having the entire control of the procedures for backing up and accessing our data, in a long term perspective.

This first experience led us to consider an home made solution with a dedicated tool for storing and sharing secondary datasets together with all analysis results obtained from these datasets in the frame of our multi-omics projects. For that, we worked with TailorDev (https://tailordev.fr/), a French company which is well recognized to apply best practices for software development and promote Open Science in research entities. By following our specifications, the TailorDev developers created the Pixel Web App. We intentionally decided to exclude data analysis tools from the application, and thus retain all advantages of existing solutions like Galaxy (but also all the other bioinformatics tools).

**In our laboratory, the Pixel Web App holds a strategic position in the data management required in multi-omics projets: at the starting point, but also at the final point of all new data exploration.**

We tried to underline this point with several sentences in the main text:
- "*The Pixel Web App does not provide any computational programs to analyze the data. Still, it allows helps to rapidly integrate explore and mine existing results and thus holds a strategic position in the management of research data*" (Summary, lines 32 to 34, page 2)
- "*The Pixel Web App does not provide any computational programs to analyse the data. Still, it allows to explore existing results in a laboratory and to rapidly combine them for further investigations (using for instance the Galaxy platform or any other data analysis tool).*" (Discussion, lines 308 to 311, page 11)

*1. Title is misleading. "Digital lab assistant" can mean anything. "Multi-omics": Pixel is made for a very specific type of omics data. There are many types of omics data that Pixel does not support, e.g. protein-protein interaction data. Similarly, the term data integration is misleading, as it usually implies the utility of ontologies, standard vocabularies, URI to integrate data - none of which are used in Pixel as far as I can tell. I suggest to use a different title, e.g.: "Pixel - a content management platform for quantitative omics data".*

The reviewer is right. The title we proposed was confusing. We thank him for his proposal for an alternative title and we agree with: "*Pixel - a content management platform for quantitative omics data*" (see main text, page 1). Also we removed in the main text, all references to the notion of "data integration". We understand that this can be confusing for the reader and leading to potential misinterpretations of the Pixel Web App functionalities

*2. Galaxy comparison. The latest version of Galaxy provides tags and group tags (https://docs.galaxyproject.org/en/release_18.09/releases/18.09_announce.html). Primary data can be tagged, and tags are propagated all the way to secondary data in Galaxy histories/ libraries. Galaxy has tools to filter, join and visualise tables. I like to know what the USP of Pixel is compared to Galaxy?*

This question raised by the reviewer is of great interest. He is absolutely right and we hope that our explanations (see above) are clear enough. We believe that Galaxy is an extremely powerful tool (actually we use the European instance very often in our research but also teaching activities). Our idea with the Pixel Web App was rather to answer our need to store in a long term perspective, all the data and analyses carried out by our team. We have the idea that over the time, results obtained by one researcher can be re-used by another, or by the same researcher but in the context of another project.

## Experimental design

*The software stack used to develop and deploy the software is cutting edge. The presented use case nicely illustrates the functionalities of Pixel. First, filters are used to identify two datasets tagged with the same species and "alkaline pH". Second, genes/proteins are filtered if they contain "pathogenesis" in their description. It was not clear in Line 259-262 where the description came from.*

This our mistake for not having detailed this information in the main text. The description comes from the CGD database (http://www.candidagenome.org/). CGD information was stored in our own Pixel Web App instance, as descriptions of "omic units" referring to the pathogenic yeasts *Candida glabrata* or *Candida albicans*. It is very useful to easily display only the Omics Units which are related to a given biological question ("pathogenesis" in this case).

We added a reference to the CGD database in the main text (lines 375, page 13, legend of Figure 6).

*My concerns are:*

*3. Reproducibility. Pixel clearly helps with data/content management, but I can't figure out how it helps with reproducibility. As the authors clearly state, Pixel is not a tool for primary data analysis, it is for managing and visualising secondary data. When users upload data into Pixel, they can provide a description of the "Analysis" and "Experiment". It is very optimistic to expect users would provide tool versions and parameters that created the secondary data. I would recommend to remove any references to reproducibility.*

The reviewer is right and has perfectly understood how our Pixel Web App works. The question of the reproducibility of our analyses is central in our research activities and we needed, as researchers from different disciplines (biology, statistics, computer science) to agree on "standards", *i.e.* information that is essential to keep to reproduce any analyses, any time, by anyone. Having a fully reproducible workflow for data analysis (from raw data to figures) is a difficult task. The Pixel Web App does not enforce reproducibility (in term of program versioning, workflow development, etc.). However, it will support reproducible practises and it help us do much better than we did before.

We added in the text "help" and "support" reproducibility, to prevent misinterpretation by the readers.

*4. Metadata and Tags. The success of Pixel's search depends on the richness and consistency of user-provided tags and metadata. Two users could choose different tags to describe the same dataset. How does Pixel cope with situations when different terms are used to describe the similar data, or with spelling mistakes?*

The remark of the reviewer is very interesting. We conceived the tag system to be flexible and dynamic. Also, Pixel's scope of work in the research team up to the laboratory. We expect Pixel's users to discuss and exchange before assigning tags to a dataset. However, it is very easy to rename and re-assign tags.

*5. Use case. It would be good if the author's can illustrate why doing the same task in Galaxy would be more difficult: tag search, text search, join three tables and histogram visualisation.*

As stated earlier, Pixel Web App is a data management solution, whereas Galaxy is a tool to perform in depth analysis of the data. Pixel and Galaxy cannot be compared, they are complementary tools in a cycle of data analyses (datasets are explored and exported from Pixel, analyse in Galaxy and the obtained results can be re-imported in Pixel for latter exploration, export, etc.).

*6. Excel. Despite its universality it is notorious for identifier mangling (https://genomebiology.biomedcentral.com/articles/10.1186/s13059-016-1044-7). I would like to learn what safeguards there are against this.*

Thank you for this remark. The choice of Excel has been extensively debated between us (we are a team composed of biologists, statisticians and computer scientists). We are aware of Excel's ability to mangle biological identifier such as protein or gene names. The use of Excel spreadsheet to enter Pixel Sets has been mimic to what is done is the Gene Expression Omnibus (GEO) database. However, in the import step, all Pixel names are checked against an annotated database.

## Validity of the findings

*Minor technical issues:*
*I got this following the README:*
*- 'make bootsrap' (first step):*
*- django.db.utils.ProgrammingError: relation "core_omicsarea" does not exist*
*Went away after re-running the command*
*- Then I got: /bin/sh: yarn: command not found*
*- after installing 'yarn', it went fine*
*The step 'make dev', last message from 'make dev' is 'Creating pixel-dev_node_1 ... done'. At this point the server seems started, but the stdout isn't clear on this.*
*After this step, I can see a login page at http://127.0.0.1:8000, but I have no clue what to do from there.*
*There are requirements like Yarn, which the documentation should mention.*
*The documentation should say something about what to do once the application is deployed (is there a default user? Should I create it) and it should mention first steps I can do as user.*
*Moreover, regarding the step 'or run the Django development server and the css build watcher separately', it's not clear what to do as initial steps in either case (clone and cd).*

The production installation procedure was redefined in this document available on the Github repository: https://github.com/Candihub/pixel/blob/master/docs-install/how-to-install.md

This procedure takes into account our extensive use of Docker images (all available on the Docker Hub registry) and explains the role of each container (database, web application and reverse proxy). Accordingly, the document presents a 10 lines "deployment script" to use for a production server. In this case, usage of Docker instances solves all dependency issues (like yarn), plus populate the database (by playing a Django database migration playbook) and finally expose the Django application behind a web reverse proxy.  Finally, this document details the way to launch the Django administration script in order to create the first user (superuser).

# Reviewer: Laurent Gatto

## Basic reporting

*Denecker et al. describe the Pixel infrastructure, a digital lab assistant to manage, annotate and integrate datasets produced with a variety of omics technologies. In the first part of the manuscript, the authors describe the software in terms of underlying technology and design, and provide and overview of its usage. They then provide a short use case, leading to the identification of units potentially involved in the pathogenicity of Candida glabrata under alkaline stress using two omics datasets. The article is generally well written and conveys a good idea of the software's capabilities and the underlying technologies and development model.*
*In terms of spelling, I have two comments*
*- Line 79, possibly replace Importation with Import.*
*- Line 255, 'are search_ed_'*

We corrected these spelling mistakes. We thank the reviewer for its careful reading of our paper.

## Experimental design

*Here, I will comment on the software and its features:*

*- I struggled to install the software locally. Given the open development model used by the team, I will directly address this in a GitHub issue.*

We thank the reviewer for having reporting to us issues during the installation step. Everything should be running smoothly now.

*- As the authors acknowledge, installation of the software would require dedicated support, even if its use, later, is very intuitive. Given my own issues to install and test Pixel, and that potential users would want to try it out before installing it locally, it would be useful if a public instance could be used as test bench.*

Yes. Reviewer can now test the Pixel Web App with the following credentials:
http://pixel.candihub.eu
- Login: visitor
- Password: pixelwebapp

Note that this access allows visitors to test the functionalities to explore and export Pixel Sets (data importation and modification are not allowed). We use this Pixel Web App instance to share transcriptomics and proteomics datasets produced in our laboratory and which are presently under consideration for publication. We thank the reviewers for not communicating about them.

*- I think further clarification or examples of what the Value and QS can be used for would be helpful. It becomes clear, later, in the example, that they can be used to store fold changes and p-values, but can any other data be used such as, for example classification or clustering results? Is there thus a limitation to 2 quantitative values per unit? How would one deal with a case where the experiment compared more than two groups and more fold changes and p-values are to be reported?*

This reviewer's comment is quite very interesting and important for potential users of the Pixel Web App. We designed the application around the concept of "Pixel Set". By definition a Pixel Set comprises multiple "Omics Units" (several thousands in case of omics datasets) to which two quantitative values are associated, *i.e.* the Value and the quality score (QS).

Therefore, the Pixel Set is a very flexible format to organize data. A clustering result (as suggested by the reviewer) could be thus stored in the form of a Pixel Set in which classified objects (e.g. proteins) would be associated with their cluster number (this would be the "Value"). In such situation, the quality scores might be not filled, or filled with any quality criteria that describe the clustering relevance for the classified objects. However, it should be emphasized that this way to store clustering results in the Pixel Web App would be atypical, using qualitative information (the cluster number) as quantitative information (the Value field of a pixel). Even if the Pixel Web App can support this, it is clearly not an ideal situation. Combining in a more relevant way, both qualitative information (clustering results for instance) and quantitative information (fold changes for instance) would be an interesting evolution of the application which is actually under consideration in our lab.

In case where the experiment compared more than two groups and more fold changes and p-values, as many as needed Pixel Sets will be created (by the user) and input in the Pixel Web App.

*- Are the tags used to document Pixel sets free text, or is there some controlled vocabulary that will prevent different users to use different synonyms for the same term?*

So far, tags are free text to let users as much freedom as possible in Pixel Sets annotation. As raised by reviewer 1, the same Pixel Sets could by annotated with different tags. However, tag management is very intuitive is the Django admin interface and we expect users to discuss with each other while working on the same Pixel Web App instance.

*- On lines 188-190, I think there's a need for more information about 'user notebooks'. Are these dynamically executed scripts, or only provided as a documentation? Is this optional or mandatory?*

These "user notebooks" could be R or Python scripts used to prepare the data. They are optional. The reviewer's comment regarding the "user notebook" is very important and actually it was discussed within our team. If the idea to dynamically execute analysis scripts would have been great, it append to be very complex to implement. Also, we wanted all researchers (including biologists with only a few programming skills) to be able to store their results in the Pixel Web App without too many restrictions. This is why the "user notebook" is only documentation in the current version of the Pixel Web App. However, imposing more contraignant standards is a possible (and certainly desirable) natural evolution of the application.

*- I would suggest to link to documentation in docs and docs-install in README.md file.*

We thank the reviewer for this remark. README.md primary intended use was for "developer", the "doc-install.md" is for deployment on a production server. Documents will be merged on the Github repository.

*- Is it possible to export and update Pixel sets?*

Currently this manipulation is not possible. But you can remove an entire Pixel Set and resubmit it necessary with updated annotations and associated metadata.

*- On line 315, at the end of the discussion, the authors explain that a private web app contains 20000 pixels. Following the definition of a pixel being one line/entry in a pixel set, this seems very low.*

We apologize for the misunderstanding. We have several Pixel instances in our laboratory. This is the one which is now available for external visitors (https://pixel.candihub.eu), that comprises around 20000 pixels.

We deleted from the main text the reference to our private Web App (lines 315 to 318, page 12).

## Validity of the findings

*Here, I will focus on the use case.*

*My main concern is that Pixel Set A zip file is missing and that the Pixel Set B isn't available at all. Hence, even if I had managed to install the software, I wouldn't be able to test it by reproducing the use case presented in the manuscript.*

We apology for this. User case Pixel Set A zip file were submitted as supplementary data files. Note that to import Pixel Sets from this ZIP files, it is necessary to have a database in which the genomic entries for the studied species (here Candida glabrata) are pre-registered. We provide a script for this task on GitHub :
https://github.com/Candihub/pixel/blob/701ee39795d572536929b864bb31bc97e8733fd3/bin/load-all-entries

*I also find the use case very simplistic, or lacking details, to make a convincing case. For example, the integration, based on common identifiers (unit names), which I assume, need to be mapped accordingly prior to importing the data into Pixel, is trivial. How would Pixel deal with metabolomics data, where liking between gene/protein names and metabolites isn't as direct? A more elaborated use case, that would illustrate a data analysis cycle as presented in figure 1C, would be useful.*

*In the use case, units matching the 'pathogenicity' term are searched for. How were these units annotated originally? This seems to be crucial to be able to perform a search and identify the units of interest. How could a user identify units matching a specific GO term, for example?*

*In the use case, the authors use p-values as an example. Are these not adjusted p-values, or shouldn't they be? This, in my opinion, also reduces the relevance of the use case.*

We agree that the case study presented in this article is very simple. But it is emblematic of our daily use of the Pixel Web App (searching for a set of Pixel Sets, checking for particular genes of interest, etc.). Of course, a more tricky case study would be very interesting and the reviewer is absolutely right. But such a case study would required the application of external analysis tools (Galaxy for

instance) and considering the clarifications we had to perform on that point (see our previous answers), we think that it could be misleading for the readers.

## *Comments for the author*

*To summarise, my main concerns are*
*- being able to test the software and either reproduce the use case, or at least use test data provided by the developers;*
*- the use case could be improved to make a stronger case in favour of Pixel.*

We wrote this to article to share the Pixel Web App source code. Of course, the Pixel application has been developed to meet our needs and they can be very specific, intimately connected to the way we work together. Our aim with this article is to communicate and discuss the work we made in terms of $i$) software conceptualization and $ii$) code source production. We hope this can be interesting for other researchers in the field. Again, we would like the thank the the reviewers for their careful reading and their relevant comments.